# Neuroplasticity Elicited by Modified Pharyngeal Electrical Stimulation: A Pilot Study

**DOI:** 10.3390/brainsci13010119

**Published:** 2023-01-10

**Authors:** Xue Zhang, Xiaolu Wang, Yunxiao Liang, Yilong Shan, Rong Song, Xin Li, Zulin Dou, Hongmei Wen

**Affiliations:** 1Department of Rehabilitation Medicine, The Third Affiliated Hospital, Sun Yat-sen University, Guangzhou 510630, China; 2Key Laboratory of Sensing Technology and Biomedical Instrument of Guangdong Province, School of Biomedical Engineering, Sun Yat-sen University, Guangzhou 510006, China; 3Department of Dermatology, The Third Affiliated Hospital, Sun Yat-sen University, Guangzhou 510630, China

**Keywords:** pharyngeal electrical stimulation, swallow, dysphagia, neuroplasticity, functional near-infrared spectroscopy

## Abstract

Modified pharyngeal electrical stimulation (mPES) is a novel therapeutic method for patients with neurogenic dysphagia and tracheostomy. However, the underlying neural mechanisms are still unclear. This study aims to investigate the impact of mPES on swallowing-related neural networks and involuntary swallowing frequency using functional near-infrared spectroscopy (fNIRS). 20 healthy volunteers participated in this study, including two separate experimental paradigms. Experiment 1: Immediate effect observation, 20 participants (10 female; mean age 47.65 ± 10.48) were delivered with real and sham mPES in random order for 8 repetitions. fNIRS signals were collected during the whole period of Experiments 1. Swallowing frequency was assessed during sham/real mPES. Experiment 2: Prolonged effect observation, 7 out of the 20 participants (4 female; mean age 49.71 ± 6.26) completed real mPES for 5 sessions (1 session/day). 13 of the 20 participants withdrew for personal reasons. Hemodynamic changes were recorded by fNIRS on day 1 and 5. Results show that mPES evoked cortical activation over a distributed network in bilateral primary somatosensory, primary motor, somatosensory association cortex, pre-motor and supplementary motor area, dorsolateral prefrontal cortex, Broca’s area, and supramarginal gyrus part of Wernicke’s area. Meanwhile, the increased frequency of involuntary swallowing was associated with decreased frontopolar activation (frontopolar cortex: Channel 6, *p* = 0.024, r = −0.529; Channel 23, *p* = 0.019, r = −0.545). Furthermore, after five days of mPES, decreased cortical activations were observed in the right dorsolateral prefrontal and supramarginal gyrus part of Wernicke’s area, and left frontopolar and M1 areas. Overall, these results might suggest that mPES could elicit changes in neuroplasticity that could reorganize the swallowing-related neural network and increase involuntary swallow frequency.

## 1. Introduction

Swallowing is a complex sensory-motor task that requires coordination of the cortical and subcortical areas, the brainstem and more than 25 pairs of muscles to transport food and liquids from the lips to the stomach safely and efficiently [1,2,3]. Neurogenic dysphagia is defined as impaired safety and/or efficiency of the swallowing process [4,5] secondary to neurological diseases associated with increased aspiration pneumonia, malnutrition, cost of care and mortality [6,7,8]. However, the quality and level of most of the clinical evidence supporting effective treatment are currently limited [9,10].

Pharyngeal electrical stimulation (PES) is a promising therapeutic neurostimulation tool for patients with neurogenic dysphagia and tracheostomy, which can induce swallowing reflex by the bipolar-ring electrodes of a nasogastric tube [11]. The parameters of PES are as follows: square wave, pulse width 0.2 ms, frequency 5 Hz, 10 min per day, 3 consecutive days [11,12,13]. Previous studies [12,13,14,15,16,17,18,19] utilizing PES for neurogenic dysphagia treatment in unselected patients showed conflicting results. In some trials [12,13,15,16,18,19,20], PES was associated with alleviated clinical dysphagia. However, in dysphagic patients with subacute stroke, PES was safe but could not improve residual aspiration and dysphagia [14,17]. This may be related to most of the patients recruited having mild dysphagia that tended to recover spontaneously. Moreover, the stimulus intensity was lower than those in other PES studies. Subsequently, a prolonged regimen of PES was explored in a patient with severe infratentorial dysphagia and proved to be potentially effective [21]. However, there are a lack of studies on prolonged regimens in patients with severe dysphagia.

Modified pharyngeal electrical stimulation (mPES) was modified from PES [22]. The parameters of mPES are as follows: mixed triangular and square wave, pulse width 10 ms, frequency 5 Hz, 10 min per day. The location of mPES is guided by EMG. Our preliminary study indicated that mPES could improve swallowing function without severe adverse reactions in severe chronic neurogenic dysphagia patients [22,23]. The underlying neural mechanism of immediate and prolonged effect induced by mPES is unclear yet, but probably similar to that of PES. Several studies [20,24,25] have shown that PES can enhance the reorganization of the swallowing-related motor cortex, activate the corticobulbar pathway and increase substance P in saliva.

Functional near-infrared spectroscopy (fNIRS) is a non-invasive and non-ionizing technique for brain function research by monitoring cerebral hemodynamic changes. Compared with other neuroimaging tools, fNIRS shows several advantages in portability, safety, low cost and satisfactory spatiotemporal resolution, enabling real-time monitoring of brain activity under natural situations [26,27,28]. Simultaneous functional magnetic resonance imaging (fMRI)-fNIRS research revealed that fNIRS measurement of deoxygenated hemoglobin (HbR) is highly correlated with fMRI measurement of blood oxygen level-dependent (BOLD), suggesting that fNIRS could be an appropriate substitute for fMRI. Moreover, fNIRS measurement of oxygenated hemoglobin (HbO) and total hemoglobin (HbT) can provide more comprehensive insight into brain function [29,30]. In the present study, we applied fNIRS in investigating mPES-induced neural activity in healthy participants. Two separate experimental paradigms were conducted to examine the immediate and prolonged neural effect elicited by mPES. Regarding the fact that HbO represents higher sensitivity and robustness to task-related stimuli [31], here we only focused on the differences in cerebral cortical activation among the experimental conditions, using coefficient data for HbO. We hypothesized that mPES could induce cortical activation in some part of the swallowing-related brain regions, including the prefrontal, primary somatosensory, primary motor, insula, supramarginal gyrus and cingulate cortex, and elicit neuroplasticity after prolonged intervention.

## 2. Materials and Methods

### 2.1. Participants

20 right-handed healthy volunteers participated in this study. The required sample size is 20, which was determined by calculating the minimum sample size of paired *t*-test with 0.6 effect size and 0.75 power at *α* = 0.05 (two-tailed) using G*Power soft-ware (G*Power version 3.1, Heinrich-Heine-Universität, Düsseldorf, Belgium, Germany). The exclusion criteria included: a history of alimentary tract disease, pulmonary disease, neurological disease, musculoskeletal disorders, speech disorders, voice problems, or masticating or swallowing difficulties. Two separate experiments were con-ducted for the study. 20 participants completed Experiment 1. After a washout period of at least 30 days, 7 out of the 20 participants completed Experiment 2. There is no se-lection for the participants who took part in Experiment 2, and the 13 participants withdrew only for personal reasons. The study was approved by the Ethics Committee of the Third Affiliated Hospital, Sun Yat-sen University ([2021]02-259-01). Written in-formed consent was obtained from all participants. Clinical trial registration: Chinese Clinical Trial Register (ChiCTR2100054548).

### 2.2. mPES

The mPES device (ZIMMER, Neu-Ulm, Germany) [22] contained a portable EMG device, a control panel for regulating parameters and a tube for electromyography (EMG) signals recording and stimulus delivery. There were two pairs of ring electrodes (reference and active electrodes) with a distance of 8.8 cm on the tube. The stimulus was a mixed waveform (triangular and square waves) with a frequency of 5 Hz and a pulse width of 10 ms. 

The tube was inserted into the pharyngeal cavity through the nose with the guidance of EMG amplitude. When the pharyngeal EMG declined steadily below 20 µV for the first time, the participant was instructed to swallow saliva. Then, the EMG amplitude increased above 20 µV rapidly, indicating good contact between the active ring electrode and the hypopharyngeal mucosa, specifically the piriform sinus [22]. 

The stimulus current intensity (CI) was measured twice. The initial CI was 0.5 mA, then it was gradually increased by 0.5 mA. The perception threshold (PT) was the lowest CI at which the participant could feel the stimulation. The maximum tolerance threshold (MTT) was the CI at which the participant felt pain or discomfort and did not want the CI to increase further. The optimal CI was calculated as PT+(0.75×[MTT−PT]). The stimulation duration was 10 min/day. The current was applied during mPES, and no current was applied during the sham intervention.

### 2.3. fNIRS Data Acquisition

A continuous-wave fNIRS system (Nirscan 24 × 24, DanYang HuiChuang Medical Equipment Co. Ltd., Danyang, Jiangsu, China) was utilized to record cerebral hemodynamic signals. The system used three wavelengths (730 nm, 808 nm, and 850 nm) working at a sample rate of 11 Hz. A total of 63 channels were created, with 24 sources and 24 detectors with a source–detector distance of 3 cm. Figure 1A shows the placement of optodes and channels. The placement of the fNIRS cap was validated by centering the specific mark at Cz, based on the international 10–20 system [32]. The setup of fNIRS cap was conducted by the same experimenter in this study to minimize the influence of experimenter effect. After setting the fNIRS cap, the participant was instructed to grasp with their right hand; activation in channels corresponding to the contralateral motor cortex could be an indicator of good positioning of the fNIRS cap.

A standard head phantom with a head circumference of 58 cm was used for spatial registration of the channels; this was regarded as the averaged head for all participants [33]. After putting the fNIRS cap on the head phantom, the anatomical locations of each of the optodes in relation to 5 reference landmarks, including nasion (Nz), inion (Iz), right preauricular point (AR), left preauricular point (AL) and top center (Cz), were collected using a Patriot 3D Digitizer (Polhemus, Colchester, Vermont, USA). Then, the spatial registration of channels to Montreal Neurological Institute (MNI) space [34] and the corresponding Broadmann area based on the Talairach atlas [35] was obtained using the built-in functions of NIRS-SPM software [36], and the positions of channels were projected to the brain, as shown in Figure 1B. 

### 2.4. Experimental Procedures

The experiment was conducted in a quiet and dim room to avoid distractions. To avoid the influence of biorhythm, all participants were treated and assessed around 6 pm, and they were instructed to refrain from eating or drinking in case of emesis during the mPES.

The experimental setup is shown in Figure 2A. The participants sat comfortably on a chair. After setting the mPES electrodes and fNIRS cap, they were informed that they should relax their arms on their thighs with their eyes closed. Throughout the experiment, they were informed that they should relax without exerting any mental effort or falling asleep.

#### 2.4.1. Experiment 1: The Immediate Effect Observation

The purpose of this experiment section was to observe the immediate effect induced by mPES. Figure 2B shows the diagrammatic representation of the experimental procedure. Participants passively received real/sham mPES during the experiment. The stimuli were presented in blocks (25 s) followed by 25 s rest. Each condition (real and sham mPES) was repeated 8 times and presented alternately, while the order of real or sham mPES at the onset of the stimulus was randomized to balance the sequential effect. Before the start of the trial, the experimenter would explain to the participants in advance that the real or sham mPES would appear randomly at the onset of the stimulus. At the onset of real/sham mPES, the experimenter would tap on the mPES control panel, creating noticeable sound, which could be an indication of the onset of stimulation. fNIRS signals were collected during the total duration of Experiment 1. The number of swallows during the first mPES and sham condition for 25 s was counted by observing the movements of the thyroid cartilage [24].

#### 2.4.2. Experiment 2: The Prolonged Effect Observation

In this additional experimental section, we tried to investigate the changes in mPES-related cortical activation after prolonged interventions. Figure 2C shows the diagrammatic representation of the experimental procedure. Participants received mPES for 5 sessions (1 session/day). fNIRS signals were recorded during mPES on the first and fifth days. For each session, real mPES were delivered in blocks (repeated 10 times). Each block lasted 30 s, followed by 30 s rest. The duration of blocks and the number of repeats were both increased compared with Experiment 1, thereby strengthening the effect of the intervention. Prior to and post mPES, participants were required to maintain a resting state for 7 min.

### 2.5. Data Analysis

Pre-processing of fNIRS data was performed using Homer2 (v2.8) software [37]. Signal quality was visually checked by inspecting the power spectrum of raw data for each channel and individual, and channels without the presence of prominent cardiac components (~1 Hz) were removed (the excluded channels are listed in Appendix A). Since participants were required to close their eyes throughout the experiment, channels of the visual cortex (Ch: 42, 43, 58, 59, 60, 61, 62, 63) were also excluded, leaving 55 channels for further analysis. The pre-processing procedures were as follows (see Appendix A). Firstly, raw data was converted to optical density. And channel-wise motion artifacts were identified using the function hmrMotionArtifactByChannel, by which marking the sample points exceeding the threshold of the given amplitude (AMPthresh = 5) and standard deviation (STDEVthresh = 10) within a given period (tMask = 3) as motion artifacts. Then, motion artifacts were corrected using spline [38] and wavelet [39] methods. A bandpass filter of 0.01–0.1 Hz was then applied to remove confounding factors of periodic and mPES-irrelevant physiological noises. Based on modified Beer-Lambert’s law (MBLL), we calculated changes in HbO and HbR concentration. Here, we focused our analysis for HbO considering its high sensitivity and signal-to-noise ratio [40,41]. Finally, the preprocessed data were exported to NIRS-KIT (v1.3.2) [42] software, and mPES-evoked cerebral cortical activation (*β* coefficient) relative to baseline was calculated based on the general linear model (GLM).

Statistical analyses were performed using MATLAB (R2020b). Kolmogorov-Smirnov test was performed to test if the data followed normal distribution. Specifically, for Experiment 1, we performed paired *t*-test (normal distributed) and Wilcoxon rank sum test (non-normal distributed) to compare the difference between real and sham mPES evoked cortical activation and involuntary swallowing frequency. For Experiment 2, changes in cortical activation evoked by real mPES between the first and fifth day were compared by paired *t*-test (normal distributed) or Wilcoxon rank sum test (non-normal distributed). Multiple comparisons were corrected across channels using the Benjamini-Hochberg false discovery rate (FDR) method. Correlations be-tween involuntary swallowing frequency and cortical activity were analyzed by partial Pearson correlation with age and gender as covariates. The threshold of statistical significance was set at *p* < 0.05.

## 3. Results

32 participants were assessed for eligibility, 8 participants did not meet the eligibility criteria and 4 participants declined to be involved in the trial. Finally, 20 participants (10 female, 10 male) aged 23 to 64 years (mean = 47.65, SD = 10.48) participated in Experiment 1, and 7 participants (4 female, 3 male) aged 37 to 55 years (mean = 49.71, SD = 6.26) participated in Experiment 2. The 7 participants participated in both Experiment 1 and 2 with a washout period of at least 30 days. Flow diagram is shown in Figure 3.

Compared with the sham mPES, mPES showed a significant effect on the frequency of involuntary swallows (times/30 s) (Figure 4A). By comparing the real mPES condition with the sham one, we recognized various cortical areas significantly activated by real mPES. Figure 4B shows the result of mPES versus sham activation for HbO, involving the supramarginal gyrus part of Wernicke’s area (BA-40; Ch: 2, 48), dorsolateral prefrontal cortex (DLPFC) (BA-9, 46; Ch: 3, 11, 18, 20, 24, 25, 29, 34, 35, 39), Broca’s area (BA-44, 45; Ch: 4, 12, 19, 26), primary somatosensory (S1) (BA-1, 2, 3; Ch: 14, 15, 44, 45), somatosensory association cortex (SAC) (BA-5, 7; Ch: 16, 53, 56), pre-supplementary motor area (Pre-SMA) (BA-6; Ch: 27, 36, 38, 40, 46, 47) and primary motor area (M1) (BA-4; Ch: 30, 57). Cortical activations in these areas significantly increased and showed left lateralization in DLPFC. We further evaluated the relationship between swallowing behavior and cortical activity by estimating partial Pearson’s correlation between times of involuntary swallow and beta coefficient, with age and gender as covariates. This estimation was performed for each channel, and we found significant correlations in some part of frontopolar cortex (FPC) (BA-10; Ch: 6, 23) at uncorrected level (Figure 4C).

Concerning the prolonged effect, we compared cortical activation between post- (the fifth session) and pre- (the first session) mPES. As shown in Figure 5, significant decrease was found in left M1 (BA-4; Ch: 56, 57), right supramarginal gyrus part of Wernicke’s area (BA-40; Ch: 48), left frontopolar cortex (BA-10; Ch: 9), and right dorsolateral prefrontal cortex (DLPFC) (BA-46; Ch: 18, 20) for HbO. Changes in mPES-related cortical activation of these channels were shown in Figure 5. Additionally, the PT of the 20 participants were all 0.5 mA, of which the MTT of 18 participants was 1mA, and MTT of 2 participants was 1.5 mA. No severe adverse effect has been reported during mPES, except two participants experienced transient nausea in the first mPES intubation.

## 4. Discussion

Our results suggested that mPES has a potential impact that could increase involuntary swallow frequency, lead to changes in neuroplasticity and enhance neural efficiency of swallowing-related networks after prolonged intervention sessions. 

Neural activity during swallowing has been widely reported in the past few decades. There are two forms of swallowing: voluntary and involuntary swallowing. Similar cortical activation patterns have been observed during voluntary swallowing, involving brain regions of the prefrontal, S1, M1, insula, supramarginal gyrus and cingulate cortex [43,44,45]. For involuntary swallowing, however, cortical activation was predominantly in the bilateral S1 and M1 [45]. In the present study, we recognized mPES-evoked cortical brain regions of the bilateral S1, M1, SAC, Pre-SMA, DLPFC, Broca’s area and supramarginal gyrus part of Wernicke’s area. These aforementioned areas play an essential role in the initiation, execution, modulation and control of swallowing motor behavior [46,47,48,49], suggesting that mPES-related neural effect is not limited to involuntary swallowing but is involved in the regulation and compensation of the entire swallowing network. This is likely the primary neural mechanism contributing to swallowing function recovery, consistent with previous PES studies utilizing fMRI and magnetoencephalography (MEG) [24,50]. In line with previous findings, we also found that individual with higher executive capability (i.e., higher involuntary swallowing frequency) was associated with weaker activation in the FPC, suggesting higher neural efficiency for individual with higher swallowing performance [51].

Concerning the prolonged effect, cortical activation in left M1, left FPC, right supramarginal gyrus part of Wernicke’s area and right DLPFC decreased after five days’ mPES. The inhibited activation might associate with neuroplastic changes and the higher efficiency of cortical connections in these regions [52]. This result is consistent with a previous finding [24], reflecting a stimulus-induced increase in swallowing processing efficiency. More specifically, the right M1, FPC and right supramarginal gyrus part of Wernicke’s area have been suggested to play a crucial role in the pharyngeal stimuli and dysfunction [24,53,54]. Meanwhile, the right FPC might also be as-sociated with the prospective memory task cued by aversive stimuli (e.g., mPES) [55,56]. Finally, according to the resource conservation theory [57,58], the deactivation of DLPFC was for the preserved mental effort during prolonged challenging task maintenance (e.g., mPES-induced swallowing). Nevertheless, due to the small sample size, we do not deny the possibility of neuroplastic changes in other brain regions associated with swallowing. 

This study has some limitations. First, the inherent limitation of the fNIRS technique only allowed us to detect neural activity on the superficial layer of the cerebral cortex. Inner structures such as the brainstem and cerebellar cannot be detected. Multi-modal measurements of concurrent fMRI-fNIRS or EEG-fNIRS [59,60,61] have recently attracted some interest, enabling a comprehensive brain function analysis. Systemic interference from the scalp is another problem that presents in long separation measurements in the present study. Short separation channels of less than 1 cm could be an effective method to reduce such contamination [62,63]; these would be included in future studies. Additionally, the positioning of channels is a challenging issue presented in fNIRS studies. Previous studies have recommended personalized optimal montages to guarantee accurate investigation of specific cortical regions [64]. However, considering the high requirement of time and cost, we compromised and used a head phantom for spatial registration; this may have lowered the specificity of measurements. Concerning the influence of skin and hair properties, we did not conduct a biased selection of participants. Some studies would select participants with hair shorter than 1 cm (or a bald head) and light skin to ensure high signal quality, but these findings may lack generalizability. Despite this, the influence of delayed hemodynamic response related to the mPES was also neglected. Future studies explicitly exploring this issue are needed to better understand the underlying neural mechanism of mPES, thereby enlightening the development of future neurorehabilitation approaches. 

Another limitation is that the current study is conducted on healthy adults with a relatively small sample size without a control group. In Experiment 2, only 7 participants completed the prolonged effect observation, which constrained the statistical power. The results of decreased activation after prolonged mPES and the correlation of cortical activation and swallowing frequency did not survive after multiple comparison corrections, so we can only make relatively conservative interpretations. Nevertheless, these results are noteworthy that they revealed a potential role of mPES in eliciting neuroplasticity, which informs further investigations on patients. Future randomized controlled mPES studies with a large sample size of dysphagia patients should be carried out to consolidate our findings. 

## 5. Conclusions

In conclusion, the current findings provide preliminary evidence of mPES-induced increased involuntary swallow frequency, activated extended swallowing-related cortical regions and enhanced efficiency of the swallowing process after a prolonged intervention. These results demonstrated that mPES might reorganize the swallowing-related neural network, suggesting that mPES might be a promising therapeutic neurostimulation method for severe neurogenic dysphagia. Further prospective studies with large patient cohorts are needed to validate the results.

## Figures and Tables

**Figure 1 brainsci-13-00119-f001:**
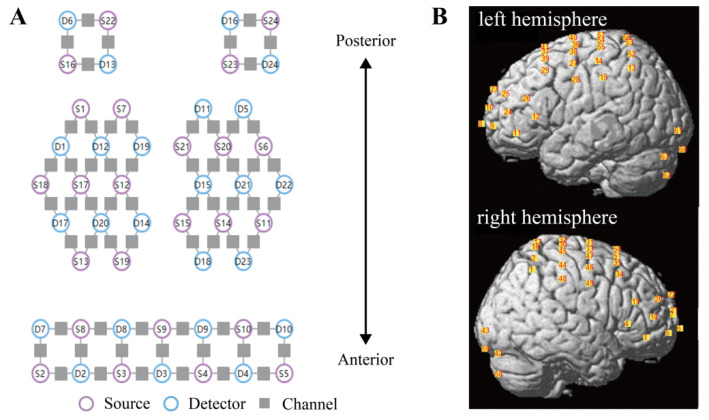
Configuration of fNIRS optodes. (**A**) The layout of sources (purple circles), detectors (blue circles), and channels (gray squares). A total of 24 sources and 24 detectors were placed with a source–detector distance of 3 cm, creating 63 channels. (**B**) Spatial registration of channels. Positions of channels on the brain are shown in the left (top figure) and right (bottom figure) hemispheres. fNIRS functional near-infrared spectroscopy.

**Figure 2 brainsci-13-00119-f002:**
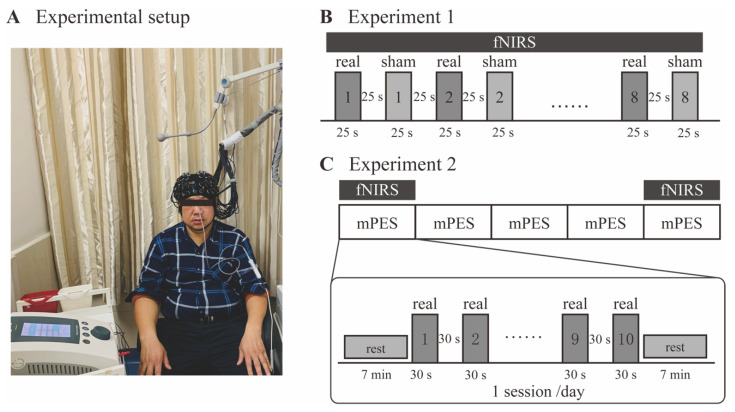
Experimental design. (**A**) A participant sitting on the chair after setting of the mPES electrodes and the fNIRS cap. (**B**) The procedure of Experiment 1. Real and sham mPES were delivered alternately in blocks (repeated 8 times), while the order of real and sham at the onset of mPES was randomized. Each block lasted 25 s, followed by 25 s rest. fNIRS signals were recorded concurrently. (**C**) The procedure of Experiment 2. Participants received mPES for 5 sessions (1 session/day). fNIRS signals were recorded during mPES on the 1st and 5th days. For each session, real mPES were delivered in blocks (repeated 10 times). Each block lasted 30 s, followed by 30 s rest. mPES modified pharyngeal electrical stimulation; fNIRS functional near-infrared spectroscopy.

**Figure 3 brainsci-13-00119-f003:**
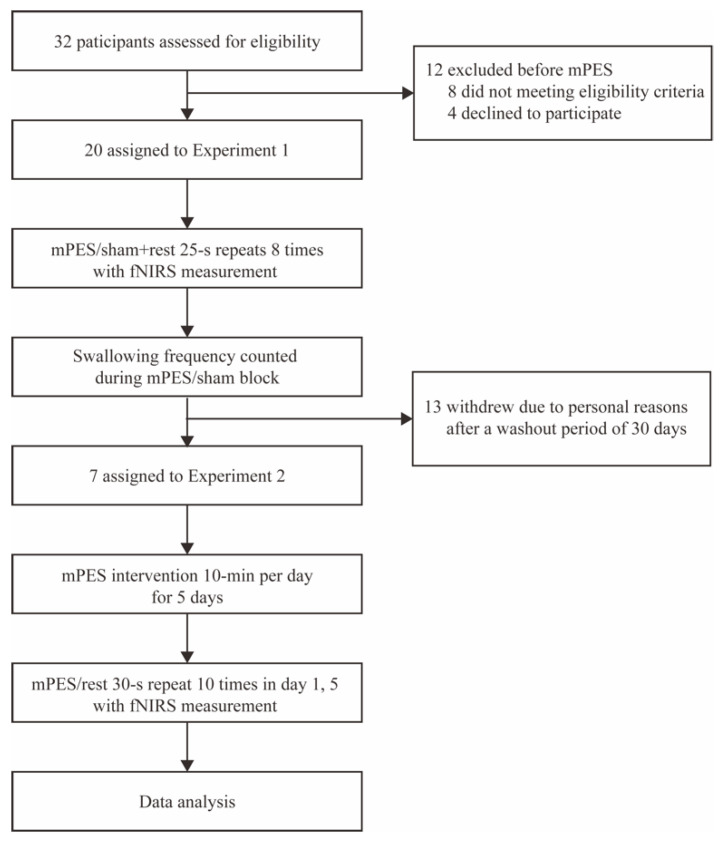
Flow diagram of the trial. mPES modified pharyngeal electrical stimulation; fNIRS functional near-infrared spectroscopy.

**Figure 4 brainsci-13-00119-f004:**
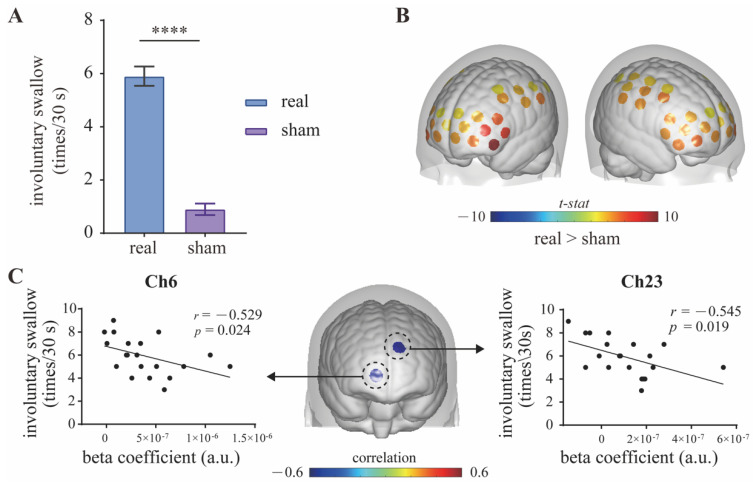
Contrast between real and sham mPES. (**A**) The number of involuntary swallows induced by real and sham mPES. *n* = 20, **** *p* < 0.0001, Wilcoxon test. (**B**) Real mPES versus sham mPES activation for HbO. The colored regions denote cortical areas that were significantly activated (paired *t*-test, *p* < 0.05, FDR corrected). Statistical results refer to Table 1. (**C**) Cortical activity (beta coefficient) correlates with behavior (involuntary swallow frequency). Partial Pearson’s correlation with age and gender as covariates, *n* = 20, uncorrected. mPES modified pharyngeal electrical stimulation.

**Figure 5 brainsci-13-00119-f005:**
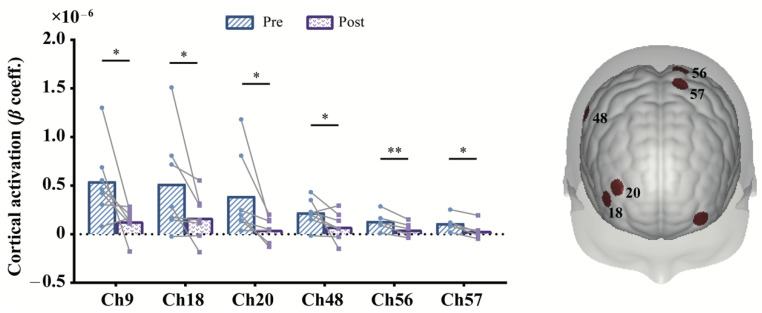
Contrast of post- and pre-prolonged mPES for HbO. The colored regions on the brain denote the locations of these channels. * *p* < 0.05, ** *p* < 0.01, uncorrected. Statistical results refer to Table 2. mPES modified pharyngeal electrical stimulation.

**Table 1 brainsci-13-00119-t001:** Real versus sham mPES-related cortical activation during immediate mPES.

BA Region	Ch#	MNI [X, Y, Z]	*β* Mean Difference (95% CI) × 10^−7^ (Real–Sham)	df	t	*p*
**Left hemisphere**						
BA-40	13	[−56, −54, 51]	0.435 (0.079, 0.792)	19	2.557	0.030
BA-9, 46	11	[−52, 44, −6]	4.465 (3.322, 5.609)	16	8.279	<0.001
	24	[−45, 53, 12]	3.280 (2.285, 4.274)	17	6.958	<0.001
	25	[−35, 55, 28]	1.500 (0.743, 2.256)	19	4.151	0.001
	29	[−47, 21, 48]	1.004 (0.540, 1.469)	19	4.527	0.001
	39	[−39, 20, 58]	0.536 (0.111, 0.960)	19	2.64	0.028
BA-45	12	[−57, 29, 7]	3.490 (2.033, 4.946)	14	5.139	0.001
	26	[−51, 36, 24]	2.398 (1.605, 3.192)	17	6.378	<0.001
BA-10	8	[−14, 74, 1]	3.198 (1.735, 4.662)	18	4.591	0.001
	9	[−36, 65, −1]	3.129 (2.109, 4.149)	18	6.446	<0.001
	10	[−24, 68, 16]	1.918 (0.893, 2.943)	19	3.916	0.002
	23	[−11, 64, 31]	0.917 (0.239, 1.595)	19	2.831	0.019
BA-1,2,3	14	[−56, −27, 56]	0.568 (0.109, 1.028)	18	2.6	0.029
	15	[−66, −30, 41]	1.276 (0.760, 1.791)	19	5.178	<0.001
BA-6	27	[−52, −4, 55]	1.214 (0.511, 1.916)	18	3.631	0.004
**Right hemisphere**						
BA-40	2	[42, −54, 63]	0.429 (0.083, 0.774)	19	2.594	0.029
	48	[68, −31, 44]	2.070 (1.093, 3.046)	18	4.454	0.001
BA-9, 46	3	[54, 44, −9]	4.422 (2.630, 6.215)	18	5.183	<0.001
	18	[50, 51, 10]	3.332 (2.006, 4.659)	18	5.277	<0.001
	20	[40, 54, 26]	1.871 (0.951, 2.792)	19	4.254	0.001
	34	[50, 19, 48]	1.951 (1.187, 2.716)	19	5.346	<0.001
	35	[41, 19, 58]	0.858 (0.264, 1.453)	19	3.022	0.013
BA-45	4	[60, 28, 4]	3.382 (1.937, 4.826)	16	4.963	0.001
	19	[56, 33, 23]	2.541 (1.493, 3.590)	18	5.093	<0.001
BA-10	5	[40, 64, −3]	4.183 (2.601, 5.764)	18	5.557	<0.001
	6	[15, 73, 0]	3.588 (1.967, 5.209)	19	4.633	0.001
	7	[29, 68, 13]	2.482 (1.346, 3.619)	19	4.573	0.001
	21	[2, 68, 15]	2.291 (0.872, 3.710)	19	3.379	0.006
	22	[16, 66, 29]	1.119 (0.173, 2.065)	19	2.475	0.035
BA-1,2,3	44	[57, −31, 57]	1.065 (0.388, 1.741)	18	3.305	0.008
	45	[45, −30, 68]	0.648 (0.260, 1.037)	19	3.491	0.005
BA-6	46	[55, −6, 54]	1.982 (1.091, 2.874)	19	4.653	0.001
	47	[43, −6, 65]	0.783 (0.277, 1.288)	19	3.237	0.008

Notes: Two-tailed paired *t*-test, FDR corrected. Ch# = Channel (similarly hereinafter). mPES modified pharyngeal electrical stimulation.

**Table 2 brainsci-13-00119-t002:** Post-versus pre-mPES-related cortical activation after prolonged mPES.

BA Region	Ch#	MNI [X, Y, Z]	*β* Mean Difference (95% CI) × 10^−7^ (Post–Pre)	df	t	*p*
**Left hemisphere**						
BA-10	9	[−36, 65, −1]	−4.075 (−7.947, −0.204)	6		0.047
BA-4	56	[−18, −50, 76]	−0.824 (−1.257, −0.391)	4	−5.286	0.006
	57	[−17, −27, 78]	−0.722 (−1.044, −0.401)	4	−6.238	0.003
**Right hemisphere**						
BA-46	18	[50, 51, 10]	−3.421 (−7.405, 0.563)	6		0.031
	20	[40, 54, 26]	−3.416 (−6.812, −0.02)	6		0.031
BA-40	48	[68, −31, 44]	−1.420 (−3.134, 0.294)	6		0.047

Notes: Two-tailed paired *t*-test, uncorrected. mPES modified pharyngeal electrical stimulation. Ch# channel.

## Data Availability

Data are available for consultation upon request to the corresponding author.

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
