# Peer review of "Neuroplasticity Elicited by Modified Pharyngeal Electrical Stimulation: A Pilot Study"

_brainsci, 2023, doi:10.3390/brainsci13010119_

Round 1

Reviewer 1 Report

Main message of the article

Neuroplasticity elicited by modified pharyngeal electrical stimulation: a pilot study (brainsci-2062155) (Review)

The article “Neuroplasticity elicited by modified pharyngeal electrical stimulation: a pilot study” by Zhang and colleagues used functional near- infrared spectroscopy (fNIRS) to explore the neural basis of modified pharyngeal electrical stimulation with a sample of 20 participants for Experiment 1 and 7 participants for Experiment 2. Significant neural activation was observed after mPES in typical somatosensory and motor regions. Furthermore, involuntary swallowing was associated with decreased frontoparietal activity.

General Judgment Comments

The manuscript is clearly summarized by its title and its keywords. The Abstract needs to be clearer and to be consistent with the other sections of the manuscript (e.g., it is not clear that the authors used fNIRS in Experiment 1 and 13 participants withdrew from the study). It is not clear how the authors decided the sufficient sample size for the study. However, a sample 20 participants seems quite a small number to have enough statistical power for the analysis. Furthermore, 13 participants withdrew and poses a bias for the selection of the participants enrolled in Experiment 2. Results in the manuscript are not reported following the standard format. Figures are informative, but Figure 3 needs to be more specific in describing the use of fNIRS during Experiment 1. Another further issue of the study is that the second Experiment does not have a control condition, so it is not clear how the authors can link the observed results to the mPES.

Major Issues

  • -  Please include participants’ main demographic information in the Abstract.

  • -  How was the sufficient sample size computed for the study? A sample of 20 participants seems quite small, especially considering the final remaining sample (n = 7) for Experiment 2.

  • -  Please report the significant results in the Abstract by following the standard format.

  • -  In Experiment 2, the authors do not seem to have any control condition. How do they know whether the observed effects are due to the modified pharyngeal electrical stimulation (mPES) or just to any type of stimulation/experiment/amount of time?

  • -  Lines 27-29: the authors stated that “These results indicate that mPES could effectively induce a more efficient swallowing process and reorganize the swallowing-related neural networks associated with increased involuntary swallow frequency”. How did the authors reach this conclusion?

  • -  From the Abstract it is not clear that also during Experiment 1 the authors recorded neural activity with fNIRS. Please clarify.

  • -  Figure 3 is not accurate as the panel b does not mention the use of fNIRS. Please modify accordingly. In panel 2 it is not clear that the blocks with “mPES’ represents days of experiment.

  • -  From the Abstract, the fact that 13 participants withdrew did not emerge. This might be problematic as there is a selection for the people that take part in Experiment 2.

  • -  Please clarify the reasons for which correlations are uncorrected. The authors should use Bonferroni’s correction to minimize the risk of false positive results.

  • -  The hypothesis is a bit generic: what are the swallowing-related brain regions that the authors refer to?

  • -  How were motion artifacts identified in fNIRS signals? Please clarify in the manuscript.

  • -  Results: results are not reported in the main text by following the standard format.

    Minor Issues

- Line 17: “ethology”? Please clarify.

Author Response

Dear Editor and Reviewers,

Thanks very much for taking your time to review this manuscript (ID: brainsci-2062155). We really appreciate your constructive comments and suggestion. We have carefully considered the suggestions and made some changes. Please find our itemized responses in below and our revised manuscript in the re-submitted files. We hope that your comments have been addressed accurately.

The changes have been highlighted in yellow in the revised manuscript and listed in accordance with each comment below. We look forward to hearing from you regarding our submission. We would be glad to respond to any further questions and comments that you may have.

Response to Reviewer 1

The article “Neuroplasticity elicited by modified pharyngeal electrical stimulation: a pilot study” by Zhang and colleagues used functional near- infrared spectroscopy (fNIRS) to explore the neural basis of modified pharyngeal electrical stimulation with a sample of 20 participants for Experiment 1 and 7 participants for Experiment 2. Significant neural activation was observed after mPES in typical somatosensory and motor regions. Furthermore, involuntary swallowing was associated with decreased frontoparietal activity.

General Judgment Comments

The manuscript is clearly summarized by its title and its keywords. The Abstract needs to be clearer and to be consistent with the other sections of the manuscript (e.g., it is not clear that the authors used fNIRS in Experiment 1 and 13 participants withdrew from the study). It is not clear how the authors decided the sufficient sample size for the study. However, a sample 20 participants seems quite a small number to have enough statistical power for the analysis. Furthermore, 13 participants withdrew and poses a bias for the selection of the participants enrolled in Experiment 2. Results in the manuscript are not reported following the standard format. Figures are informative, but Figure 3 needs to be more specific in describing the use of fNIRS during Experiment 1. Another further issue of the study is that the second Experiment does not have a control condition, so it is not clear how the authors can link the observed results to the mPES.

Major Issues

-  Please include participants’ main demographic information in the Abstract.

Response 1: Thanks for the insightful comments, which have helped us greatly in improving our manuscript quality. The corresponding modifications have been made in the Abstract section as follows:” Experiment 1: Immediate effect observation, 20 participants (10 female; mean age 47.65± 10.48) were delivered with real and sham mPES in random order for 8 repetitions. fNIRS signals were collected during the total duration of Experiments 1. Swallowing frequency was assessed during sham/real mPES. Experiment 2: Prolonged effect observation, 7 participants (4 female; mean age 49.71± 6.26) completed real mPES for 5 sessions (1 session/day).”

-  How was the sufficient sample size computed for the study? A sample of 20 participants seems quite small, especially considering the final remaining sample (n = 7) for Experiment 2.

Response 2: We really appreciate your careful comments. We have added information about the sample size calculation as follows: “The required sample size is 20, which was determined by calculating the minimum sample size of paired t-test with 0.6 effect size and 0.75 power at α = 0.05 (two-tailed) using G*Power software.” However, because of the invasive procedure, we had difficulty recruiting healthy volunteers, and only 7 participants completed Experiment 2, which constrained the statistical power. We acknowledge that this is the major limitation in the current study. We have also addressed this limitation in the Discussion as follows:” Another limitation is that the current study is conducted on healthy adults with a relatively small sample size without a control group. In Experiment 2, only 7 participants completed the prolonged effect observation, which constrained the statistical power. The results of decreased activation after prolonged mPES and the correlation of cortical activation and swallowing frequency did not survive after multiple comparison corrections, so we can only make relatively conservative interpretations. Nevertheless, these results are noteworthy that they revealed a potential role of mPES in eliciting neuroplasticity. Future randomized controlled mPES studies with a large sample size of dysphagia patients should be carried out to consolidate our findings.”

-  Please report the significant results in the Abstract by following the standard format.

Response 3: Thanks for your careful reminder. The significant results have been changed to “(frontopolar cortex: Channel 6, p= 0.024, r= -0.529; Channel 23, p= 0.019, r= -0.545)”.

-  In Experiment 2, the authors do not seem to have any control condition. How do they know whether the observed effects are due to the modified pharyngeal electrical stimulation (mPES) or just to any type of stimulation/experiment/amount of time?

Response 4: Thanks for your constructive comments. Because of the invasive procedure of mPES, we have difficulty in recruiting enough healthy volunteers to set up the control group. We acknowledge that this is a major limitation of the present study. We have added these to the Discussion Limitation section as follows:” Another limitation is that the current study is conducted on healthy adults with a relatively small sample size without a control group. In Experiment 2, only 7 participants completed the prolonged effect observation, which constrained the statistical power. The results of decreased activation after prolonged mPES and the correlation of cortical activation and swallowing frequency did not survive after multiple comparison corrections, so we can only make relatively conservative interpretations. Nevertheless, these results are noteworthy that they revealed a potential role of mPES in eliciting neuroplasticity. Future randomized controlled mPES studies with a large sample size of dysphagia patients should be carried out to consolidate our findings.”

-  Lines 27-29: the authors stated that “These results indicate that mPES could effectively induce a more efficient swallowing process and reorganize the swallowing-related neural networks associated with increased involuntary swallow frequency”. How did the authors reach this conclusion?

Response 5: Thanks for your insightful comment. To avoid any over-interpretation, we have revised the conclusion as follows:” Overall, these results might suggest that mPES cause changes in neuroplasticity that could reorganize swallowing-related neural networks and increase the frequency of involuntary swallowing.”

-  From the Abstract it is not clear that also during Experiment 1 the authors recorded neural activity with fNIRS. Please clarify.

Response 6: Thanks for your patience and kind reminder. These has been added in the Abstract and Method section as follows:” fNIRS signals were collected during the whole period of Experiments 1.”

-  Figure 3 is not accurate as the panel b does not mention the use of fNIRS. Please modify accordingly. In panel 2 it is not clear that the blocks with “mPES’ represents days of experiment.

Response 7: Thanks for your careful and kind reminder, and the modified figure is shown in Figure 3 below.

-  From the Abstract, the fact that 13 participants withdrew did not emerge. This might be problematic as there is a selection for the people that take part in Experiment 2.

Response 8: Thanks for your valuable comment. No selective enrollment was performed in Experiment 2. The 13 participants withdrew due to personal reasons. We have revised the information to the Abstract as follows: “7 out of the 20 participants (4 female; mean age 49.71± 6.26) completed real mPES for 5 sessions (1 session/day). 13 of the 20 participants withdrew for personal reasons.” We also added statement in the Method section to clarify this issue: “Two separate experiments were conducted for the study. 20 participants completed Experiment 1. After a washout period of at least 30 days, 7 out of the 20 participants completed Experiment 2. There is no selection for the participants who took part in Experiment 2, and the 13 participants withdrew only for personal reasons.”

-  Please clarify the reasons for which correlations are uncorrected. The authors should use Bonferroni’s correction to minimize the risk of false positive results.

Response 9: Thanks for your insightful comment. We agree that multiple comparison correction should be conducted to reduce the risk of false positive rate. In the current study, we performed FDR correction (commonly used in fNIRS studies) for multiple comparison correction across the 55 channels (8 out of the 63 channels covering the visual areas were excluded for all the subsequent analysis). We have reported the corrected results in Figure 4B. However, due to the small sample size, we didn’t observe significant results for the correlation analysis in Experiment 1 and post- versus pre- comparisons in Experiment 2 after FDR corrections. Therefore, we could only report the statistical results with lower statistical power in Figure 4C and Figure 5. Nevertheless, these findings are noteworthy. The results presented in Figure 4C is consistent with prior evidence that individual with better motor capability (i.e., higher involuntary swallowing frequency) is associated with weaker activation, while the results presented in Figure 5 revealed a potential role of mPES in eliciting neuroplasiciticy. The presentation of uncorrected results is also a common practice in neuroscience studies when the statistical results didn’t survive after multiple comparison corrections [1,2].

Nonetheless, we acknowledge that this is a major limitation in the current study. To avoid over-interpretations, we have added more explanations on this issue in Discussion. “In line with previous findings, we also found that individual with higher executive capability (i.e., higher involuntary swallowing frequency) was associated with weaker activation in the FPC, suggesting higher neural efficiency for individual with higher swallowing performance.” “The results of decreased activation after prolonged mPES and the correlation of cortical activation and swallowing frequency did not survive after multiple comparison corrections, so we can only make relatively conservative interpretations. Nevertheless, these results are noteworthy that they revealed a potential role of mPES in eliciting neuroplasticity. Future randomized controlled mPES studies with a large sample size of dysphagia patients should be carried out to consolidate our findings.”

Reference:

[1] Pan, Y., Dikker, S., Goldstein, P., Zhu, Y., Yang, C., & Hu, Y. (2020). Instructor-learner brain coupling discriminates between instructional approaches and predicts learning. Neuroimage, 211, 116657.

[2] Cléry, J. C., Schaeffer, D. J., Hori, Y., Gilbert, K. M., Hayrynen, L. K., Gati, J. S., . . . Everling, S. (2020). Looming and receding visual networks in awake marmosets investigated with fMRI. Neuroimage, 215, 116815.

-  The hypothesis is a bit generic: what are the swallowing-related brain regions that the authors refer to?

Response 10: Thanks for your careful and kind suggestion. The swallowing-related brain regions refer to the prefrontal, primary somatosensory, primary motor, insula, supramarginal gyrus, and cingulate cortex. According to the reviewers' suggestion, we have modified our working hypothesis as follows:” We hypothesized that mPES could induce cortical activation in some part of the swallowing-related brain regions, including the prefrontal, primary somatosensory, primary motor, insula, supramarginal gyrus, and cingulate cortex, and elicit neuroplasticity after the prolonged intervention.”

-  How were motion artifacts identified in fNIRS signals? Please clarify in the manuscript.

Response 11: Thanks for the insightful comment. The relevant elaboration is incorporated into the Method part as follows:”And channel-wise motion artifacts were identified using the function hmrMotionArtifactByChannel, by which marking the sample points exceeding the threshold of the given amplitude (AMPthresh = 5) and standard deviation (STDEVthresh = 10) within a given period (tMask = 3) as motion artifacts. Then, motion artifacts were corrected using spline [38]and wavelet [39] methods.”

Reference:

[38] Scholkmann, F.; Spichtig, S.; Muehlemann, T.; Wolf, M. How to detect and reduce movement artifacts in near-infrared imaging using moving standard deviation and spline interpolation. Physiol Meas 2010, 31, 649-662, doi:10.1088/0967-3334/31/5/004.

[39] Robertson, F.C.; Douglas, T.S.; Meintjes, E.M. Motion artifact removal for functional near infrared spectroscopy: a comparison of methods. IEEE Trans Biomed Eng 2010, 57, 1377-1387, doi:10.1109/tbme.2009.2038667.

-  Results: results are not reported in the main text by following the standard format.

Response 12: Thanks for your insightful comment. We have moved the result of recruitment and the corresponding figure to the result item and present the results following the format required.

Minor Issues

- Line 17: “ethology”? Please clarify.

Response 13: Thanks for your patience and kind reminder. We have revised “ethology” to “involuntary swallowing frequency”.

Reviewer 2 Report

The proposed study is very interesting for the clinical and scientific community. However, some points need to be improved and detailed.

In the materials and methods section, why were healthy participants with such a wide age range included? Part of item 2.1, which refers to the result of recruitment, could be transferred to the result section, so as not to cause confusion in interpretation, since in reality 32 subjects were recruited and after a selection process, explained in figure 1, only 20 subjects were included. Figure 1 could also be transferred to the results item.

In the experimental procedure, experiment 1, what is the guarantee or orientation given to the volunteer so that the swallowing for the analysis always occurs, even if it is involuntary? Was any additional evaluation carried out to verify if the volunteer was able to perceive the difference when he was in the sham or real condition?

As there was a large age range among the volunteers, you could have analyzed this data as a covariate, or plotted this result, to see if there is any influence on the age issue, for future work with patients.

Author Response

Dear Editor and Reviewers,

Thanks very much for taking your time to review this manuscript (ID: brainsci-2062155). We really appreciate your constructive comments and suggestion. We have carefully considered the suggestions and made some changes. Please find our itemized responses in below and our revised manuscript in the re-submitted files. We hope that your comments have been addressed accurately.

The changes have been highlighted in yellow in the revised manuscript and listed in accordance with each comment below. We look forward to hearing from you regarding our submission. We would be glad to respond to any further questions and comments that you may have.

Response to Reviewer 2

The proposed study is very interesting for the clinical and scientific community. However, some points need to be improved and detailed.

In the materials and methods section, why were healthy participants with such a wide age range included? Part of item 2.1, which refers to the result of recruitment, could be transferred to the result section, so as not to cause confusion in interpretation, since in reality 32 subjects were recruited and after a selection process, explained in figure 1, only 20 subjects were included. Figure 1 could also be transferred to the results item.

Response 1: Thanks for your constructive comments. With respect to the wide age range, the recruitment of healthy participants was difficult due to the invasive procedure in our trial. Meanwhile, we have analyzed age as a covariate when performing correlation analysis and carried out partial correlation analysis. While in the comparative analysis of true and false stimuli, because it is a repeated measurement design, a paired t-test is performed. In our future work with patients, we would pay more concern on this topic.

Considering Part of item 2.1, we agree that the result of recruitment and figure 1 should be transferred to the result item, and the corresponding modification has been highlighted in our revised manuscript.

In the experimental procedure, experiment 1, what is the guarantee or orientation given to the volunteer so that the swallowing for the analysis always occurs, even if it is involuntary? Was any additional evaluation carried out to verify if the volunteer was able to perceive the difference when he was in the sham or real condition?

Response 2: Thanks for your meaningful comments. In fact, the trial was designed based on the existing literature [1] and our early observations [2,3]. In our pilot study of the mPES procedure, by observing the laryngo-hyoid complex elevation and flexible endoscopic evaluation of swallowing (FEES), we found that sham stimulation mainly induced foreign body sensation in the pharynx, which prompted the volunteers to “swallow” the foreign body. While high-frequency involuntary swallowing was observed during real mPES in all the volunteers. Both methods are accurate, whereas observing the laryngo-hyoid complex elevation is easier to operate and inexpensive.

Reference:

  1. Tsukano, H., Taniguchi, H., Hori, K., Tsujimura, T., Nakamura, Y., & Inoue, M. (2012). Individual-dependent effects of pharyngeal electrical stimulation on swallowing in healthy humans. Physiology & behavior, 106(2), 218–223. https://doi.org/10.1016/j.physbeh.2012.02.007
  2. Zhang, X.; Wang, X.; Dou, Z.; Wen, H. A novel approach to severe chronic neurogenic dysphagia using pharyngeal sensory electrical stimulation: a case report. Am J Phys Med Rehabil 2022, doi:10.1097/phm.0000000000002116.
  3. Zhang, X.; Liang, Y.; Wang, X.; Shan, Y.; Xie, M.; Li, C.; Hong, J.; Chen, J.; Wan, G.; Zhang, Y.; et al. Effect of Modified Pharyngeal Electrical Stimulation on Patients with Severe Chronic Neurogenic Dysphagia: A Single-Arm Prospective Study. Dysphagia 2022, doi:10.1007/s00455-022-10536-z.

As there was a large age range among the volunteers, you could have analyzed this data as a covariate, or plotted this result, to see if there is any influence on the age issue, for future work with patients.

Response 3: Thanks for your insightful comments. We agree that age should be analyzed as a covariate. In fact, we have considered the factors of age and gender as covariates when performing partial Pearson correlation analysis. In the comparative analysis of real and sham mPES, because it is a within-subject design, to our knowledge, paired t-test is appropriate. We will also carefully address the age and gender factors in subsequent trials involving patients.

Reviewer 3 Report

In this paper, the authors have used fNIRS to investigate the impact of mPES on the swallowing-related neural mechanism. The topic of the paper is timely. Overall the paper is well-structured and written. The followings are the comments;

1) Did the author focus on any post-experiment questionnaires about the experiment or paradigm from the patient? Any opinion has been taken from the participants on whether they feel any fatigue or difficulty in performing the paradigm.

2) "A standard head phantom with head circumference of 58 cm was used for spatial registration of the channels, which was regarded as the averaged head for all participants." Kindly provide a reference for this strong statement.

3) In experiment 2: Why fNIRS data was not obtained and analyzed after each day? 

Author Response

Dear Editor and Reviewers,

Thanks very much for taking your time to review this manuscript (ID: brainsci-2062155). We really appreciate your constructive comments and suggestion. We have carefully considered the suggestions and made some changes. Please find our itemized responses in below and our revised manuscript in the re-submitted files. We hope that your comments have been addressed accurately.

The changes have been highlighted in yellow in the revised manuscript and listed in accordance with each comment below. We look forward to hearing from you regarding our submission. We would be glad to respond to any further questions and comments that you may have.

Response to Reviewer 3

In this paper, the authors have used fNIRS to investigate the impact of mPES on the swallowing-related neural mechanism. The topic of the paper is timely. Overall the paper is well-structured and written. The followings are the comments;

1)    Did the author focus on any post-experiment questionnaires about the experiment or paradigm from the patient? Any opinion has been taken from the participants on whether they feel any fatigue or difficulty in performing the paradigm.

Response 1: Thanks for your constructive comment. In fact, we have asked the participants whether they felt any discomfort and recorded the information in the clinical record form (CRF) post-intervention in our study. There were only two participants experienced transient nausea in the first mPES intubation (Result section). In our pretest about the procedure of this study, we found that participants would feel fatigue and restless in 10 minutes. Therefore, the total duration of mPES in Experiments 1 and 2 has been set to around 10 minutes to reduce discomfort. In our future research with large sample size, well-designed standard detailed post-experiment questionnaires would be adapted to gain more comprehensive information.

2)    "A standard head phantom with head circumference of 58 cm was used for spatial registration of the channels, which was regarded as the averaged head for all participants." Kindly provide a reference for this strong statement.

Response 2: Thanks for the kind comment. The reference has been added as follows: “Fishburn, F. A., Ludlum, R. S., Vaidya, C. J., & Medvedev, A. V. (2019). Temporal Derivative Distribution Repair (TDDR): A motion correction method for fNIRS. Neuroimage, 184, 171-179. doi:10.1016/j.neuroimage.2018.09.025”. This study also used a head phantom for spatial registration of the channels.

3) In experiment 2: Why fNIRS data was not obtained and analyzed after each day?

Response 3: Thanks for your insightful comment. Since our study aims to explore the neuroplasticity after the prolonged intervention, not the standard 3 days protocol, meanwhile accounting for the participants’ compliance. Therefore, fNIRS data was obtained and analyzed only on days 1 and 5.

Round 2

Reviewer 1 Report

I am sorry but I still believe the sample size does not offer any possibility to generalize. I have seen that other reviewers accepted the study, so I leave the choice to the editor. However, I do not recommend the authors publish the study without collecting more data.

Author Response

Thank you for your comments.

Reviewer 2 Report

The writers' revisions satisfy me, and I believe the manuscript is ready for publication.

Author Response

We are very grateful to you for your supports.